# South African consumers' perceptions of front-of-package warning labels on unhealthy foods and drinks

Makoma Bopape[1]*, Lindsey Smith Taillie[2], Tamryn Frank[3], Nandita Murukutla[4], Trish Cotter[4], Luyanda Majija[4], Rina Swart[5]

1 Department of Human Nutrition and Dietetics, Faculty of Health Sciences, University of Limpopo, Limpopo, South Africa, 2 Carolina Population Center and Department of Nutrition, Gillings School of Global Public Health, University of North Carolina at Chapel Hill, Chapel Hill, NC, United States of America, 3 School of Public Health, Faculty of Community and Health Sciences, University of the Western Cape, Cape Town, South Africa, 4 Vital Strategies, New York, NY, United States of America, 5 Department of Dietetics and Nutrition, Faculty of Community and Health Sciences, University of the Western Cape, Cape Town, South Africa

☯ These authors contributed equally to this work.
* makoma.bopape@ul.ac.za

**Data Availability Statement:** All focus groups transcriptions are available on Figshare (https://figshare.com/s/ace2f699638ab4b293e0).

## Abstract

Front-of-package labeling (FOPL) is a policy tool that helps consumers to make informed food choices. South Africa has not yet implemented this labeling system. The aim of this study was therefore to explore adult South African consumers' perceptions of front-of-package warning labels on foods and non-alcoholic beverages (referred to as drinks in this paper) and their insights into features that could influence the effectiveness of the warning label. Using a qualitative approach, the study purposively selected consumers diversified by urbanization, gender, socioeconomic status, and literacy. We collected data from a total of 113 participants through 12 focus group discussions. Data were systematically coded and divided into five themes namely, positive attitudes toward warning labels, perceived benefits of warning labels, perceived behavior modification, perceived beneficiaries of warning labels, and effective attributes of warning labels. Almost all participants from all socio-economic backgrounds were positive about warning labels, reporting that warning labels concisely and understandably educated them about the nutritional composition of foods. Other perceived advantages were that warning labels warn of health implications, are easily understandable and could benefit child health. Some participants anticipated that warning labels would reduce their purchases of unhealthy foods, while others thought the labels would have no effect on their purchasing habits. Participants found the warning labels attention grabbing and stated that they preferred a black triangle placed on a white background (referred to as a holding strap henceforth), the words "high in" and "warning" in bold and uppercase text, an exclamation mark, and an icon depicting the excessive nutrient. In South Africa warning labels may improve consumer understanding of nutrition information and assist consumers in determining the nutritional quality of packaged foods and drinks.

**Funding:** This study was funded by Bloomberg Philanthropies. The funders had no role in study design, data collection and analysis, decision to publish or preparation of the manuscript.

**Competing interests:** The authors have declared that no competing interests exist.

## Introduction

The global prevalence of overweight and obesity is high [1, 2] and South Africa is no exception [3, 4]. Worldwide no country has been able to turn around the rising numbers [2, 5], and the increase has been particularly steep in South Africa [3, 4]. The South Africa Demographic and Health Survey (2016) reports an overweight and obesity prevalence of 31% among South African men and an even higher prevalence of 68% among South African women [4]. The prevalence of noncommunicable diseases (NCDs), such as hypertension and diabetes, is increasing annually in low- and middle-to-high-income countries, including South Africa [4, 6]. Comprehensive and effective corrective measures are needed to address these trends.

Due to urbanization, access to a wide variety of ultra-processed foods and non-alcoholic beverages (henceforth referred to as drinks in this paper) has increased remarkably in the South African market since the early 1990s [7, 8]. Several reports have demonstrated a link between high consumption of ultra-processed foods and drinks and the development of obesity and NCDs [9–11]. Urbanization has resulted in a shift from traditional diets to consumption of ultra-processed foods [8, 12, 13] made from multi-ingredient formulated mixtures containing little if any whole foods, such as ready-to-eat refrigerated processed meat (known as polonies), carbonated drinks, biscuits, and breakfast cereals, and that are typically high in sodium, sugar, saturated fats, and calories [14]. In a study conducted in Soweto, South Africa, most adolescents reported a high frequency of fast food consumption per week, with sweets, crisps, and soft drinks accounting for more than 65% of the total items consumed [15]. In other studies South African adults have reported very high sodium intakes due to consumption of processed food, including ultra-processed food [4, 12].

A simple, easily understandable food labeling system could assist consumers in identifying unhealthy food options amid the wide variety of packaged products available in markets and steer consumers away from them. Simplified FOPL has a role to play in addressing these health concerns by empowering consumers with information about products' nutrient content. Front-of-package labeling (FOPL) is a practical tool that empowers consumers to make informed food choices at the points of purchase and consumption [16, 17]. Several international studies have demonstrated the usefulness of FOPL in assisting consumers to identify unhealthy food products and discouraging selection of products identified as unhealthy [18–21].

FOPL systems range from reductive labels, which mainly summarize the nutrition information in the back-of-pack Nutrition Information Panel and present it on the front of the package, for example, Guideline Daily Amounts (GDA), to interpretive labels, which evaluate the nutritional quality of food products and present the information with icons or symbols, for example, warning labels and health logos [22, 23]. Evidence indicates that interpretive food labels are more effective in directing food choices than their counterparts [16, 20, 24].

Warning labels are interpretive FOPL systems that are implemented in countries such as Chile and Israel to highlight products that are excessive in energy, saturated fats, sugar and sodium [25, 26]. This labelling system aims to discourage purchasing and overconsumption of unhealthy products by flagging products which contain excessive nutrients of concern in a simple, visible and easily understood manner [20, 27]. Highlighting nutrients associated with NCDs may increase risk perception, foster easy identification of unhealthy products, and discourage their purchasing and overconsumption [24, 28, 29]. Consumers have limited shopping time [30, 31] and warning labels that are conspicuous serve as a means to quickly identify unhealthy products within a short period [32].

Recent studies in France, Australia, Brazil and Chile indicate that, in comparison with other systems, warning labels are more effective in decreasing the intention to purchase unhealthy foods and drinks [18, 28, 33] and in helping consumers identify less healthy foods and drinks [34, 35]. Evidence from the tobacco industry also points to the potential of health warning labels to reduce use of a harmful product [36, 37]. The use of health warning labels with images alongside text on tobacco packages has been associated with increased smoking cessation [39] and attempted cessation [38]. Warning labels increase risk perception through eliciting negative emotions such as fear, discomfort and worry and thus are effective at reducing purchasing intention [32, 38, 39].

Although studies revealed the successfulness of warning labels in dissuading consumers from purchasing unhealthy products and decreasing the perceived healthfulness of unhealthy products [20, 40], no significant differences in the mean nutritional composition of purchased products were noted in certain cases [41]. Warning labels by their binary nature (warning vs no warning) and because they do not state nutrient amounts are reported to be less informative, especially among the highly educated groups [21, 42]. The nutrition facts panel at the back of the package could however be used to complement the front of pack nutrition information.

Nutrition warning labels highlight excessive nutrients of concern and often include the text "high in" or "excess," warning consumers that the levels of those nutrients are above health recommendations [25, 27]. The text is usually enclosed in familiar shapes such as a triangle or octagon associated with some form of risk and may be accompanied by text depicting danger or caution [25, 43]. Repeated exposure to familiar shapes such as the stop sign or triangle attracts consumer attention [44] and increases danger or risk awareness [43]. Deliza et al. [45] found that participants located black triangles and black octagons most quickly on product packages in comparison to other shapes and FOPL formats. According to the Communication–Human Information Processing (C–HIP) Model, warning labels that are attended to, are well understood and increase risk perception, influence behaviour and may ultimately lead to habit change [46–48]. Use of color such as black or red, which is associated with 'stop' or danger [49]; bold font, which is attention grabbing [43]; and bigger label size enhance the labels' visibility and captures attention [43]. In a study by Cabrera et al. [50] a red stop sign with the text 'excess' was associated with the least perceived healthfulness and black was reported more visible. These attributes enhance the warning labels' effectiveness in informing consumers and steering them away from unhealthy food [51]. In addition pictorial images and icons further increase visibility [52] and simplify understanding of warning labels across various sociodemographic groups, including low-socioeconomic, low literate groups and children [18, 32]. Images are easy to understand and are more easily retained in memory [53].

The current nutrition panel currently used in South Africa is on the back of the packaged, is complicated and not well understood due to the terminology and the difficulty in interpreting numbers used [30, 54]. A study by Jacobs et al. [55] revealed that South African consumers previously expressed a need for an FOPL system that communicates nutrition information in a simple manner. South Africa as a country has not yet implemented any FOPL system The success of warning labels [18, 34, 35] in other countries created an opportunity to develop and test a a warning label that could effectively warn and modify South African consumers' purchases of unhealthy products. This study aimed to fill this gap by probing South African consumers' responses to warning labels and exploring their views on features that could enhance or diminish its effectiveness. The study explores (1) consumers' perceptions of warning labels and (2) consumers' views on design features that could influence the effectiveness of a warning label.

## Materials and methods

### Study design

The study required an in-depth understanding of participants' views of warning labels, so we followed an exploratory descriptive qualitative approach [56]. With an exploratory descriptive qualitative design, researchers gain more knowledge of a process or a situation from the affected individuals than with other designs [57]. The materials and methods followed in this study will be presented according to the Consolidated Criteria for Reporting Qualitative Research (COREQ) [58].

### Setting

An independent market research company that has not done work for the food industry in the two years prior to this project collected data in two metropolitan and two non-metropolitan areas in each of three purposefully selected provinces in South Africa: Gauteng, KwaZulu-Natal and Western Cape. The provinces represent the country's diverse socioeconomic statuses and cultural beliefs and practices.

### Sample

The market research company recruited participants through their existing database using a recruitment questionnaire developed by the research team (MB, LST, NM, TC, LM and RS) (S1 File). Data were collected from 12 focus groups of 8 to 10 members each, with a total of 113 participants taking part in the study. A sample of forty (40) participants was targeted per province to represent the various sociodemographic strata. The sample included adults primarily responsible for household food purchases purposefully selected using quotas stratified according to gender (male or female), age (18–29 years or 30–50 years), literacy (no literacy, low literacy, or literate), income (low or middle-high), urbanicity (urban or rural), and geographic location (Gauteng, KwaZulu-Natal, or Western Cape provinces) (Table 1). The focus groups were homogenous with each group consisting of participants with similar sociodemographic characteristics. We classified household income levels as low if less than R1,600 (approximately USD 100) per month and middle-high if R1,601 and above. We defined no literacy as adults with no formal schooling, low literacy as adults who had passed grades 1–6, and literate as adults who had passed grade 7 or higher.

### Ethical considerations

We obtained ethical clearance from the University of the Western Cape Biomedical Research Ethics Committee (Reference number BM18/9/13). Fieldworkers shared a letter of information with interested participants during recruitment. We explained the aims of the study and the data collection procedure to the participants, who provided their written consent for participation prior to data collection. We also obtained participants' consent to record the discussions and informed that each group would be webcasted live to a group of researchers. Discussions were conducted in the language of the participants.

### Label design

A design agency created several warning label prototypes for testing on South African consumers following a detailed design brief based on the latest literature [36, 43]. The designer was briefed to create an effective warning label using shapes, text and icons that would increase South African consumer's identification of the presence of unhealthy nutrients in foods and encourage healthy choices. The intent was also to ensure that the labels would be effective across diverse sociodemographic groups in South Africa. A committee comprised of experts in

**Table 1. Socio-demographic characteristics of participants (n = 113).**

|  | n (%) |
|---|---|
| **Gender** | |
| Male | 31 (27.4) |
| Female | 82 (72.6) |
| **Age** | |
| 18–29 years | 26 (23) |
| 30–50 years | 87 (77) |
| **Urbanicity** | |
| Urban | 63 (55.7) |
| Rural | 50 (44.2) |
| **Literacy** | |
| No literacy (no formal schooling) | 5 (4.4) |
| Low literacy (grades 1–6) | 49 (43.4) |
| Literate (grade 7 and above) | 59 (52.2) |
| **Work status** | |
| Unemployed | 62 (54.9) |
| Self-employed | 18 (15.9) |
| Employed | 20 (17.6) |
| Part-time employed | 7 (6.2) |
| Seasonal worker | 6 (5.3) |
| **Combined family monthly income** | |
| Low (R0–R1,600) | 86 (76.1) |
| Middle-high (R1,601 and above) | 27 (23.9) |

nutrition, health, health promotion, economics, communication, and media systematically rated each South African design to narrow down the range of options. Based on recommendations by the expert committee, we first tested black triangles on a white holding strap with the word "warning" (Fig 1).

Each triangle contained the text "high in" and icons depicting the nutrient of concern present, sugar, saturated fat, and/or sodium (Fig 2). The warning labels were superimposed on the front of four different food and drinks packages i.e. chips/crisps (square packet), fruit juice (bottle), yoghurt (yoghurt container) and cereal (rectangular box). A product high in all three nutrients would show three corresponding triangles with the relevant icons. We used this original warning label to evaluate consumers' perceptions of warning labels.

The second part of the study compared different design elements of the warning label (Table 2 and S1 Fig) to determine the preferred version.

We tested 32 design element options of the warning label. This included different types of icons for each nutrient, symbol shapes (triangles versus octagons), symbol colors (black versus red), holding strap colors (black versus white), warning devices (warning text only, warning text with an exclamation mark, black no warning device and red no warning device), text fonts (uppercase versus lowercase letters), and label sizes (Table 2). Other countries have similarly tested a number of prototypes [59].

## Procedure

A trained moderator (with an assistant) from the market research company used a semi-structured focus group discussion guide or moderator guide (S2 File) prepared by the research team to collect data during March 2019 and April 2019 at suitable venues located in the pre-

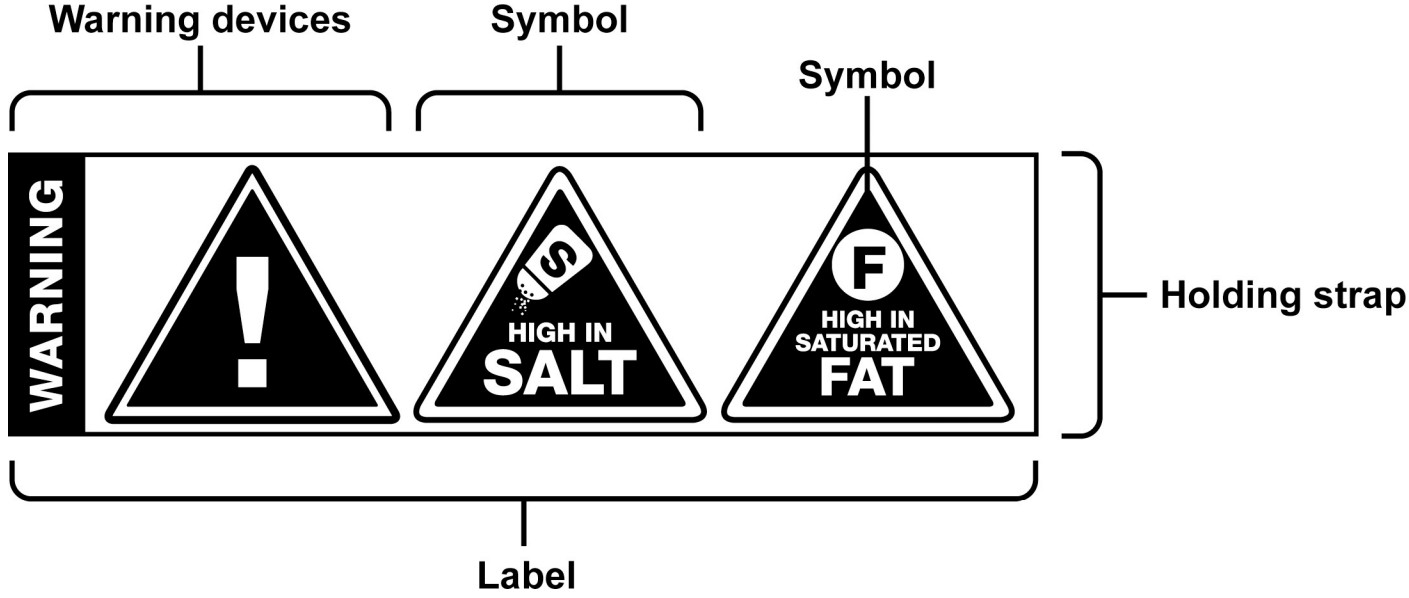

**Fig 1. Image of the warning label designed for South African consumers.**

selected areas. All the interviews were conducted by one moderator who was appointed as a researcher at the time of data collection. The moderator had extensive experience in qualitative data collection and data analysis The stimulus material (warning labels) was projected on screens and each focus group discussion was recorded with an audio-video recorder.

A graphic designer superimposed the various warning label prototypes onto real packages of crisps, cereal boxes, yogurt containers, and fruit juice bottles available on the market. Labels were placed on each package according to the excessive nutrients in those products. For example, if a product contained excessive salt and saturated fats, we placed two warning labels, one for salt and another for saturated fats, on the package. We selected products based on the 2018 Euromonitor data ranking brands according to sales. The research team selected the product that would carry the highest number of nutrient warning labels (sugar, salt and saturated fat) from the five top-selling brands in each category.

### Focus group discussion guide

We based the focus group discussion guide on instruments used in other countries. The authors of this paper collaborated to develop the guide which was piloted in two focus groups consisting of 8 to 10 participants each. The guide had two sections based on the objectives of the study. During the first section the moderator projected an image of four products, crisps,

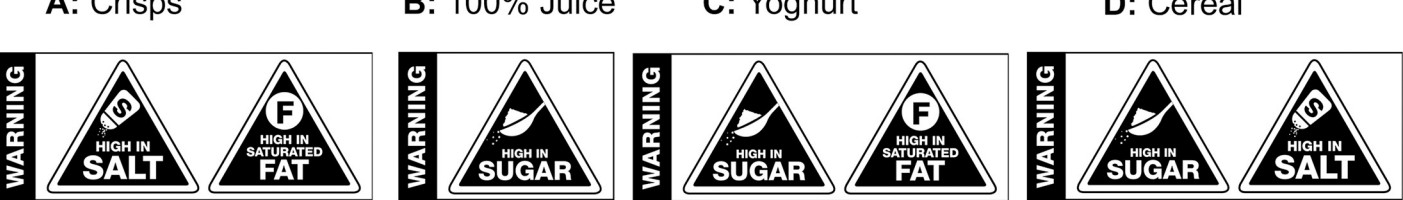

**Fig 2.** The original warning label with black and white triangles tested in the first part of the study (images from left to right: A) crisps, B)100% fruit juice, C) yoghurt and D) cereal).

**Table 2. Design elements tested.**

| Warning label element | Options tested |
|---|---|
| Icons | 12: 4 salt icons, 4 sugar icons, 4 saturated fat icons |
| Symbol shapes | 2: 1 triangle, 1 octagon |
| Symbol colors | 4: black octagon, black triangle, red octagon, red triangle |
| Holding strap colors | 2: black, white |
| Warning devices | 4: warning text only, warning text with an exclamation mark, black triangle with no warning device, red triangle with no warning device |
| Text fonts | 2: uppercase text, lowercase text |
| Label sizes | 4: 5%, 10%, 15% and 20% of the front of pack surface area |
| Placement | 2: top right corner and bottom right corner of the front of pack |

cereals, yogurts, and juices bearing the black warning label, on the screen for all participants to view and led the group discussion. The questions explored included the participants' understanding of the warning label, the label's perceived effect on purchasing habits, the label's visibility, and the label's credibility.

During the second section of the focus group discussions, the moderator projected alternatives to the original label (S1 Fig) on the screen. Participants selected the label format that was perceived as (1) attention grabbing, (2) effective as a warning against unhealthy foods and drinks, and (3) likely to influence their purchasing behaviors.

To accommodate participants' language preferences, the researchers translated the focus group guide and questionnaires, and the moderator facilitated the discussions in either English, Zulu, Xhosa, Tswana, or Sepedi. The moderator was fluent in all the languages the participants spoke. The moderator transcribed the recordings verbatim and then translated the data into English where applicable.

## Data analysis

We used a data-driven inductive thematic analysis approach to analyze the data [60]. With this approach, one member of the research team (MB) and an experienced independent qualitative researcher identified emerging codes directly from participants' responses [60, 61] and developed themes. These two researchers read and reread the transcripts to become familiar with the data [62] and separately coded the data. The two coders compared their codes and agreed on the codes and themes that best represented participants' responses.

To ensure credibility, the research team observed the focus group discussion online and held debriefing sessions with the moderator after data collection. Our researcher (MB) and the independent coder followed the same process of reading the transcripts, developing codes, and organizing data into themes based on the participants' responses. We followed the same data collection procedure with all focus groups. The authors of this article reviewed the themes and the quotations.

## Results

We extracted 5 themes and 16 subthemes from the data (Table 3 and S1 Table).

### Positive attitude toward warning labels

Participants from all socio-economic backgrounds generally had positive attitudes toward warning labels. They attributed diseases to food choices and believed warning labels would

**Table 3. Themes and subthemes.**

| Themes | Subthemes |
| --- | --- |
| Positive attitude toward warning labels | |
| Perceived benefits of warning labels | Warn of health implications |
| | Provide useful nutrition information |
| | Educational |
| | Easily understandable |
| | Benefit child health |
| | Provide succinct information |
| Perceived behavior modification | Cautiousness |
| | Indifference toward warning labels |
| Positive elements of warning labels | Visibility |
| | Color |
| | Position |
| | Text |
| | Emphasis |
| | Symbols |
| Perceived beneficiaries of warning labels | All consumers |
| | Individuals with medical conditions |

provide helpful nutrition information for food selections. One participant from a low-socio-economic background noted,

> "It is helpful;, look now we have all these ailments because we just eat anything and everything" (female, low income, low literacy, urban).

The participants also suggested that currently they lacked knowledge of nutritional quality of food products and that warning labels could enable them to make healthier food choices.

## Perceived benefits of warning labels

The focus group discussions brought up several perceived advantages of warning labels. Participants believed warning labels would warn of health implications, provide useful nutrition information, would be educational, easily understandable, benefit child health, and provide succinct information.

**Warn of health implications.** A number of participants understood that warning labels on food packages would provide information relevant to their own personal health. They were of the opinion that warning labels alert them to negative health effects associated with excessive consumption of unhealthy products. This view surfaced irrespective of socio-economic status. One participant from a rural area said, "It shows that there is some danger with the chips, if you eat them too much you might end up sick" (female, middle-high income, literate, rural). Her response conveys the label's potential to discourage excessive consumption. For her, the warning label would help people recognize the health implications of eating too much of an unhealthy product. Another participant that warning labels draw one's attention away from the palatability of the food and to other important facts about the product, "Yes, it shifts your focus from just seeing nice chips to the health hazards on them" (female, low income, low literacy, urban).

In addition, participants believed the incidences of diseases such as heart disease would decrease if warning labels were implemented: "And they will minimize the amount of chronic

diseases from the people and to those who have heart attacks, they would be easily warned (male, middle-high income, literate, rural). Another participant from a rural area also held the view that warning labels would help to prevent disease: "It will really help for preventing people from getting such ailments. I think they will really be helpful" (female, low income, no literacy, rural).

**Provide useful nutrition information.**   Warning labels state which nutrients are present in excess and thus provide useful nutrition information to consumers. One participant from a low-socioeconomic background was of the opinion that this explicit function of warning labels will make consumers aware of the nutritional quality of the food they purchase, "There are people who are not meant to take high amounts of sugar or salt and so before they buy a food item they have to check for those things, and the labels become helpful because they know exactly what the nutrient content of the product they aim to buy is" (female, low income, no literacy, urban). This suggests that because the information is available before a purchase, it could potentially influence what consumers ultimately buy.

Another participant echoed the same sentiment, adding that the information was relevant to consumers, "I think it is for us to know what kind of food are we eating and what it contains" (female, low income, low literacy, urban). A participant from an urban area noted that the message is delivered clearly due to a label's simplicity, "Because it is very simple, it is saying that there is too much salt and therefore I must not buy it" (female, middle-high income, literate, urban).

**Educational.**   The participants appreciated the labels' information, which they said was eye opening. They believed it would help them choose food more carefully and which, in turn, could help with disease management: "Personally it has changed my mindset regarding food because I now know that if I am not feeling well there is certain food that I can and cannot eat according to my health. For example, if I go to the clinic and I am diagnosed with high blood pressure then I know when I go to a store which food to pick that are healthy" (female, middle-high income, literate, rural).

Echoing that sentiment, another participant noted that the labels had enlightened her about the risks of food she had always assumed to be healthy and would help her check for the healthfulness of a product before making a purchase, "It does (help) because normally we would buy juice because it is considered healthy and now we know how to check for the levels of sugar in the juice; we now know how to check for levels of salt as well. We are well knowledgeable now (female, low income, no literacy, rural).

**Easily understandable.**   Warning labels are designed to be easily interpreted and understood by all population groups irrespective of their income levels, ethnicity and age. A woman from a middle-high income group was of the opinion that warning labels in this format would be accessible to all members of the population, the illiterate, the young, and the elderly alike. This she attributed to the icons used to depict excessive nutrients: "The labels can be easily understood by the less literate, young people, and the elderly. The sign of a spoon full of sugar, even if you do not know how to read, it makes it easy to understand this is sugar. And the salt is easy to identify since we use it when laying the table" (female, middle-high income, literate, rural). Her response suggests that all people desire access to nutrition information irrespective of social standing and labels that accommodate people's information needs could assist with that.

**Benefit child health.**   Participants perceived that warning labels would be beneficial to children's health. Showing concern for their children's well-being, they felt that knowledge about a food's nutrient content empowered them to make healthier choices for their children: "It helps us because we have children and we now know what to stop them from eating" (female, low income, no literacy, urban). This woman acknowledged that labels were a potential tool to identify products harmful to children's health.

Some participants noted that children become primary purchasers in the absence of their parents and that labels provide guidance: "A child can have R10 or R20 but now, as parents it is our duty to teach our children to stay away from food that is highly concentrated with this and that. Even if I am not there as a parent my child will know that too" (female, middle-high income, literate, urban). Another added, "When I send a child to buy something, they will see the sign and know whether this is good or not" (male, low income, no literacy, urban). The participants felt some level of comfort knowing that their children would have the means to independently judge whether food is healthful or not.

**Provide succinct information.** Warning labels only emphasize nutrients that are contained in excessive amounts without listing all the nutrition information. One urban participant appreciated the shortened and simplified presentation as it would save shopping time, "I also believe the reason why they came up with the triangle concept for the fat warning and salt is because in most cases you find that at the back they tell you about the kilojoules, so that small triangle you read fast unlike going through the whole nutritional table like the one we have now. I believe it is going to make our lives easier" (female, low income, low literacy, urban). This quote indicates that labels would simplify information in the nutrition facts panels which is a key advantage of the warning label. Consumers are interested in nutrition information but are discouraged by the long and complex nutrient list on the back of food products.

## Perceived behavior modification

This theme addresses the participants' perceptions of the effect warning labels would have on their purchasing behaviors. When they discussed it, they brought up two subthemes, cautiousness and indifference toward warning labels.

**Cautiousness.** Warning labels are designed to increase the perception of risks associated with overconsumption of unhealthy food products. Some participants hinted that the information the labels provided would prompt them to reevaluate their purchasing habits. Some participants expected that they would buy products bearing labels less often: "It raises awareness because if you know that the food contains fat and you know you are not meant to take high amounts of fat then you have the option to reduce the extent to which you buy that specific product" (female, low income, no literacy, urban). The view reflects that it is an individual's choice and that she would not necessarily stop purchasing those products but would reduce the frequency of consumption. Another participant added that she would still buy the product but would decrease her purchasing frequency: "The label will tell me but that will not necessarily stop me from buying the product. I might be influenced to buy it less often but not to entirely disuse the product, particularly if it is something I love" (female, low income, no literacy, rural).

In contrast, other participants thought they would at some point stop buying products with warning labels. A male participant indicated that, although it would be difficult, he envisaged ultimately letting go of products bearing warning labels, "The other thing is that letting go of something at once is impossible, so you reduce the amount with time and then eventually leave the product" (male, low income, no literacy, urban).

**Indifference toward warning labels.** It should be acknowledged that some participants have strong opinions about food products and some were brand loyal, declaring that labels would not have any prohibitive effect on their purchasing habits despite the dangers associated with overconsumption of some products, similar to cigarette warnings that do not convince some to stop smoking. "It is like the cigarette problem, cigarettes are written 'Dangerous: smoking can kill you,' but smokers still smoke" (female, middle-high income, literate, urban).

## Positive elements of warning labels

Participants mentioned several positive attributes that they felt could enhance the effectiveness of warning labels. They include visibility, color, position on the front of the package, text, emphasis, and symbols.

**Visibility.**   Participants appreciated that warning labels were readily visible on the pack: "When you buy the yogurt, you can easily see the label" (male, low income, no literacy, urban). Another pointed out that the black triangle design made the warning more conspicuous against the colorful product packaging, "The black sign also makes it easy to see the label" (female, low income, low literacy, rural).

**Color.**   Participants view was that black drew attention to the warning label and effectively contrasted with colorful product packages: "Even the yogurt container has bright blue and pink colors, and the black signs make you want to know what is written there" (female, middle-high income, literate, urban). The latter participant raised the interesting point that the colors raised curiosity. In the same vein, a participant from a different socioeconomic background suggested that a black and white label would raise interest in the information in the label, "It is black and white and colorful making you ask yourself what it says" (male, low income, no literacy, urban). Both participants addressed the need for warning labels to catch consumers' attention.

Some participants however seemed to prefer a red label as red is universally associated with danger. They likened red color to a red traffic light and warnings at construction sites. "A red colour would do, because we learnt at the construction sites that red means danger, if the traffic light turns red you know there is danger, it draws your attention (female, low income, low literacy, rural). Another participant added: "Even the traffic light when it signals red, it means stop or danger". (female, low income, low literacy, rural).

On the other hand there was a feeling that a red label would be too bright and ineffective when put against a red container. "That red just confuses everything because everything is red in color" (Female, middle-high income, literate, urban). Another participant added: "l think red on this package is too bright unlike on the other product, the color (black) is perfect on the package"" (female, low income, low literacy, urban). One more participant in the same group (female, low income, low literacy, urban) added: "you cannot place something with a red color on top of something with a red color".

**Position.**   Participants appreciated that warning labels were strategically placed on the front of the package, so consumers would not have to search for them: "The warning sign is in a visible place because normally for you to see the warning sign you have to turn whatever it is that you are buying to see it, but with this it is on the front, it is just there and you can see it easily" (female, middle-high income, literate, urban). This response implies that searching for nutrition information was currently inconvenient and that warning labels might be more user-friendly. Another reiterated the convenience of the position, "This label is right because it is placed in front, people will be able to notice before taking the product" (female, middle-high income, literate, rural). This participant emphasized the value of attracting the attention of customers who are not actively looking for nutrition information so they will read the label before making a purchase.

Others added another dimension to the position of warning labels. A warning label in the top right corner of a package increased its visibility compared to a label at the bottom: "And because it is at the top near the name of the product it is easy to notice it" (male, low income, no literacy, urban). Another echoed the benefit of easy access to nutrition warnings, "When one reads, they do not start at the bottom but right at the top, and so you see the one at the top much quicker than you do with the one down there" (female, low income, no literacy, urban).

**Text.**   The word "warning" particularly made an impact on our participants. They associated the text with harm linked to consumption of the product: "I would say it starts with the term 'warning,' obviously, you now know it is something that is not good, it makes us more aware" (female, middle-high income, literate, urban). This response implies that the text might contribute to the awareness that the other elements on the label initially raised. When asked how long it took them to notice the label, one participant said the word "warning" grabbed her attention and caused her to read the whole label, "It took me time until I saw the word 'warning,' then that shook me a bit (female, low income, low literacy, urban). The participant saying the text "shook" her indicates that the text elicited a strong emotional reaction.

**Emphasis.**   When asked why the labels included an exclamation mark, some participants said they did not know, while others remarked that an exclamation mark itself indicated a warning: "The exclamation mark suggests a warning" (female, low income, no literacy, urban). One added that an exclamation mark not only indicates warning but also emphasis, "[An exclamation mark] is a sign of warning and emphasis" (male, low income, no literacy, urban). He suggests that the exclamation mark, accompanying the word "warning" could intensify the impact of the label.

**Symbols.**   A combination of the triangle and icons seemed to grab participants' attention and increase their interest in the warning label: "I believe that it is easier to see based on the triangle and the icons, that of salt and sugar granules. With the products that we have that do not have the triangle sign you cannot see, when you walk into a store and when you see that triangle on the milk package obviously you want to see what is going on, are there new ingredients or . . .. I believe it can catch your eye when you walk into a store" (female, low income, low literacy, urban). Another participant from a low-socio-economic background was intrigued by the various elements on the label and stated, "It has so many signs and it will direct your eyes to the product for a longer time because you would be wondering why it has so many signs of warning" (male, low income, no literacy, urban). It sounds like the many elements made the label attention grabbing than hard to notice.

Other participants stated that they resonated more with the triangle shape than the octagon and preferred the triangle as a warning about the danger of overconsuming the labelled product. "The other one (octagon) is more visible but we prefer the triangle because we are used to it as a warning sign" (Female, Middle–High income, Literate, Urban).

## Perceived beneficiaries of warning labels

When asked to whom the warning labels were directed, some participants replied all consumers, not specific groups. Yet others understood them to be meant for individuals with medical conditions.

**All consumers.**   A participant who felt that warning labels were meant to inform all consumers said, "They are for us the consumers, to make an awareness on us that we must not just buy but check the ingredients first" (female, low income, low literacy, urban, Western Cape). As quoted in the section Benefits of warning labels above, a participant reminded us that warning labels could be relevant even to children, as they become purchasers at times, "My child will know that too".

**Individuals with medical conditions.**   However, one participant who thought warning labels were only directed to individuals with medical conditions said, "I think it works for people who suffer from ailments such as sugar diabetes so that they are able to avoid food items they are not supposed to buy" (female, low income, no literacy, rural). Another added that in the absence of medical conditions, one was free to buy as one wishes, "I do not have any health-related problems, so I can definitely buy products with lots of sugar, the one with low sugar is not tasty for me" (female, low income, no literacy, rural).

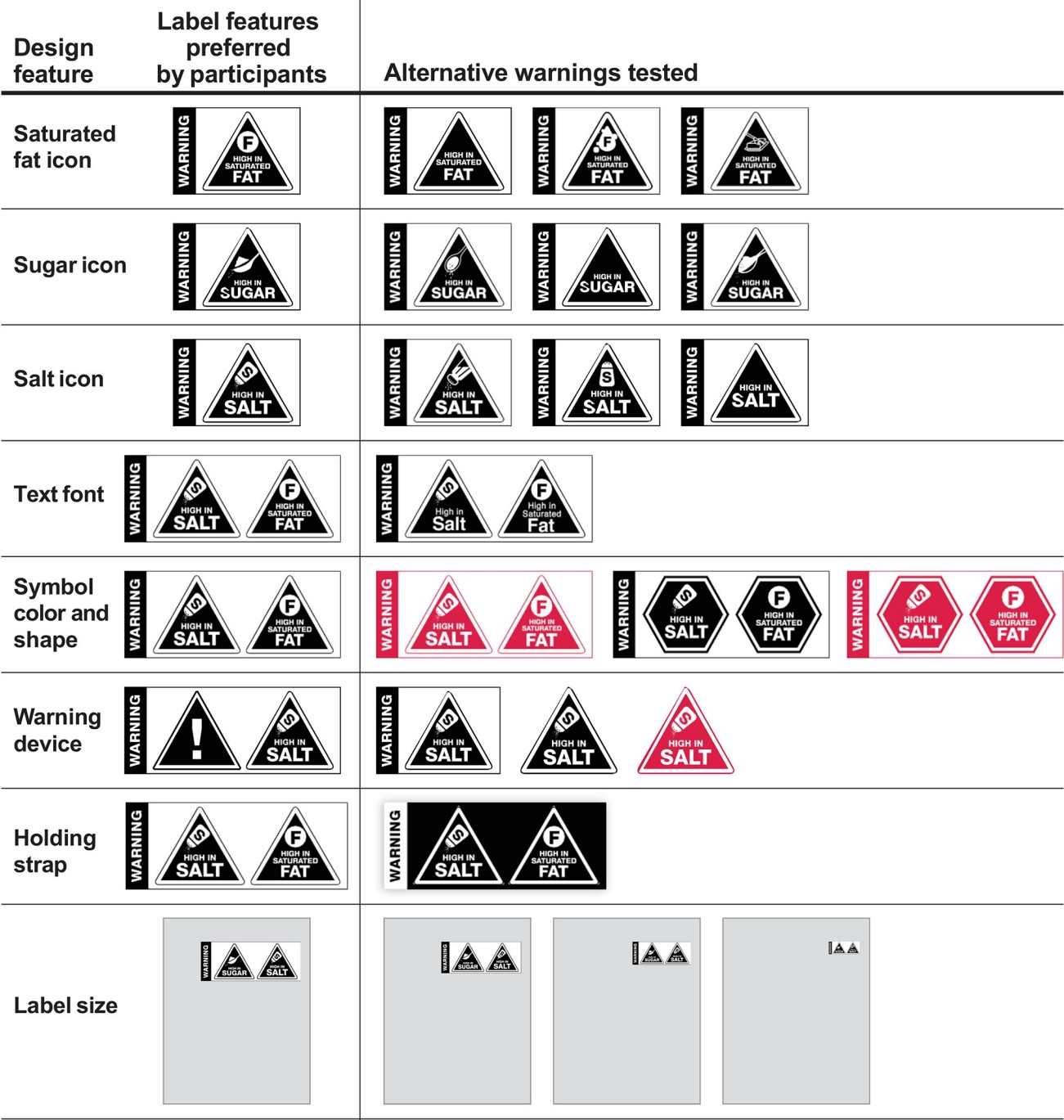

**Fig 3. Design features that appealed to participants as depicting warning.**

## Elements perceived effective as warning

Fig 2 shows the design features participants perceived as 1) attention grabbing, (2) effective as a warning against unhealthy foods and drinks, and 3) likely to influence their purchasing behaviours. They considered that a black triangle on a white background (holding strap), placement in the top right corner of the front of the package, and uppercase letters improved

the visibility of the warning label. Participants viewed a combination of the word "warning" and a triangle with an exclamation mark as signal for danger. They also preferred a larger label compared to smaller sizes (Fig 3).

## Discussion

This study found that adult South African consumers had a positive attitude toward warning labels on ultra-processed foods and drinks. Study participants felt that warning labels are easy to understand, provide important nutrition information, and could save shopping time. Our participants generally found warning labels visible and expressed that the warning labels increased risk perception. These elements are associated with the efficacy of warning labels [43]. An easily understandable and readily visible warning label attracts attention [43] and increases the likelihood of label reading, use and behavior modification [31, 43, 46, 63].Previous studies in Uruguay and Chile reported acceptance of warning labels among their participants and found the labels easy to interpret [59, 64]. Findings from quantitative studies support that consumers understand warning labels better than other label formats [27, 35, 65–67]. The advantage of easily understood warning labels is a wider coverage of consumers, even those without access to individual dietary counseling by health professionals.

This study also found a common perception that the summarized nutrition information on warning labels would easily help consumers identify unhealthy products at the point of purchase. Warning labels display only the nutrients in excess [27, 66], therefore consumers can quickly [68] and easily identify unhealthy products [18, 32]. This was supported by other participants in this study who were of the opinion that the warning labels would save them time. Consumers often have time constraints and warning labels that cut through the noise on the product packages would be beneficial [21, 69].

The use of familiar shapes [20, 52] and colors [70, 71] associated with danger increases the effectiveness of warning labels. As mandated by regulatory authorities, shapes and colors are used successfully in other industries, including the tobacco industry and segments of the food industry, to raise awareness of dangers of consuming those products [36]. However shape association with danger is culturally learned [41] and not universally interpreted in the same way [50]. In this study South African consumers perceived the black triangle in the warning label as depicting danger. A triangle is commonly used in construction sites, workplaces and road signs in South Africa to signal danger, so it is not surprising that consumers in our study in all sociodemographic groups favored it over the octagon shape. Consumers in Brazil [19] similarly related triangles with danger, but in other countries, such as Chile [59, 72] and Israel [26], consumers perceived the octagon as communicating a warning best. In the latter countries, the octagon stop sign was better understood than other shapes. These results indicate the need for each country to investigate its own context specific shape preference to improve the efficacy of the warning labels.

Although some consumers in this study, particularly those of low-socioeconomic status, perceived red as signaling danger and attention grabbing than black, black was deemed better as it contrasted with the colorful product backgrounds. Similarly, Cabrera et al. reported that black signs were easier to locate on colorful packages than red ones [50]. Color, particularly red, increases the visibility of FOPLs, increases risk perception, and influences behavior change [73]. However, visibility against the competing background is also important for warning label effectiveness [43] and a black label against a white background stood out more to our participants. In Israel, however, consumers preferred the red label [74]. Placement of the black triangle on a white background also could have improved the triangle's salience. Policy maker could further explore inclusion of a white background as a means to make the label stand out

by s. The findings of this study also confirm that consumers preferred the black triangle on the white background (i.e. holding strap) than a black triangle on a black background or without any holding strap. Chile [72] and Uruguay [75] use black and white warning labels while Israel [26] uses red and white circles.

Text and icons are among the elements that our participants thought communicated nutrition information clearly and simply. Words such as "warning," "caution," "excess," and "high in" increase risk perception and improve effectiveness [38, 43, 50]. Similarly, in this study participants pointed out that the word "warning" on labels made it clear that they were being warned about the danger associated with consumption of the product. Evidence shows that icons that summarize nutrition information are beneficial to less literate groups [32, 76, 77], which is particularly important for South Africans with lower literacy levels.

An advantage of warning labels is their potential to influence consumers' purchasing behaviors [24, 28]. Quantitative studies evaluating implementation of warning labels show that chocolate and cookie sales decreased by 8.0% and 1.2%, respectively, in Chile [78] and decreased expenditures on sweets and desserts in an online simulated environment in Uruguay [41]. In this study consumers indicated their intentions to reduce purchases of unhealthy foods with warning labels, particularly the products they like. Some even expressed an intention to stop consumption of those products altogether over time. Similarly, in a qualitative study in Brazil consumers stated they would continue consuming products with warning labels but at reduced frequency [79]. The intention to reduce consumption of ultra-processed foods and drinks is in line with the South African Food Based Dietary Guidelines, which recommend consuming fats, salt, and sugar sparingly [80]. Other participants in the current study however perceived that they would not be deterred by the warning label in line with the Health Belief Model which posits that low risk perception does not elicit behaviour change. Familiarity with products also decreases the effectiveness of the warning labels [55, 81].

An experimental study in Uruguay reported that warning labels impacted children's food choices much better than the MTL [18]. In Brazil, de Morais Sato et al. found that parents perceived that easily read and understood warning labels would help their children independently identify unhealthy food products and would increase their autonomy in healthy food choices [79]. Our participants agreed that children would benefit from simple warning labels that encourage them to make healthy choices. This observation is critical, because reducing childhood obesity by reducing their consumption of ultra-processed foods and drink is urgent. Ultra-processed foods are often marketed as convenient and palatable and front-of-package warning labels steer attention towards the unhealthiness of the products.

Our participants recognized several design features that could potentially enhance the effectiveness of warning labels, including a black triangle on a white background (holding strap), location of the warning label in the top right corner of the package, and text in uppercase letters for clear visibility. They noted that a combination of the word "warning" with a triangle containing an exclamation mark on warning labels could further effectively alert consumers to potential health risks. They also preferred a larger warning label rather than a smaller one (S1 Fig).

A strength of the study is that it considered views of consumers from diverse sociodemographic backgrounds and offered them a combination of images of foods and drinks that are perceived as healthy and unhealthy to minimize preconceived notions about the nutritional quality of the products. As with any qualitative study, the sample is not representative of the entire population and the findings cannot be interpreted statistically. Understanding labels is important for influencing behaviors, and future research should investigate the influence of the warning label on purchasing behavior in a real shopping environment.

In conclusion, our results from focus groups in South Africa suggest that a policy mandating nutrition warning labels on unhealthy packaged foods could improve consumers'

understanding of health risks and help them identify unhealthy foods and drinks. Certain design elements, such as color (black), shape (triangle), text (warning and 'High in'), use of exclamation mark and contrasting white background could enhance a label's ability to increase perception of the risks of consumption of unhealthy products.

## Supporting information

**S1 Fig. Label design elements tested.**
(PDF)

**S1 File. Participants' recruitment questionnaire for eligibility.**
(PDF)

**S2 File. Focus group discussion guide.**
(PDF)

**S1 Table. Themes, subthemes and quotes from focus groups discussions.**
(PDF)

## Acknowledgments

Special thanks to Emily Busey, graphic designer for superimposing warning labels onto food packages; to Alexey Kotov for his review of the discussion guide and for his support to the focus group discussions. We thank School of Public Health at the University of the Western Cape and the DSI/NRF CoE in Food Security (UID 91490) for administrative support.

## Author Contributions

**Conceptualization:** Makoma Bopape, Tamryn Frank, Nandita Murukutla, Trish Cotter, Luyanda Majija, Rina Swart.

**Data curation:** Makoma Bopape, Tamryn Frank.

**Formal analysis:** Makoma Bopape, Lindsey Smith Taillie.

**Funding acquisition:** Rina Swart.

**Investigation:** Nandita Murukutla.

**Methodology:** Tamryn Frank, Nandita Murukutla, Trish Cotter, Luyanda Majija, Rina Swart.

**Project administration:** Nandita Murukutla, Trish Cotter, Luyanda Majija.

**Resources:** Rina Swart.

**Supervision:** Rina Swart.

**Validation:** Nandita Murukutla.

**Writing – original draft:** Makoma Bopape.

**Writing – review & editing:** Makoma Bopape, Lindsey Smith Taillie, Tamryn Frank, Nandita Murukutla, Trish Cotter, Rina Swart.

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
