## [Decision Letter · Decision Letter 0]

21 May 2021

PONE-D-21-13158

The Editor

South African consumers' perceptions of front-of-package warning labels on unhealthy foods and drinks

PLOS ONE

Dear Dr. Bopape,

Thank you for submitting your manuscript to PLOS ONE. After careful consideration, we feel that it has merit but does not fully meet PLOS ONE’s publication criteria as it currently stands. Therefore, we invite you to submit a revised version of the manuscript that addresses the points raised during the review process.

**Associate editor's comments**:

In particular, as recommended by reviewer 2, please include a more comprehensive overview of the use of warning labels in your introduction.  For instance, you do not describe pictorial or graphic warning labels as described by Pechey et al. While this publication postdates the conduct of the focus groups, this style of warning label should nevertheless be described and the relative merits discussed.

Pechey, E., N. Clarke, E. Mantzari, A. K. M. Blackwell, K. De-Loyde, R. W. Morris, T. M. Marteau and G. J. Hollands (2020). "Image-and-text health warning labels on alcohol and food: potential effectiveness and acceptability." BMC Public Health
**20**(1): 376.

Lines 488-498 I think more could be made of the differences between countries regarding consumer shape preferences, as justification for the need to replicate this type of research locally. This finding highlights the importance of having culturally relevant shapes, as shapes are not universally interpreted in the same manner.

Please check the guidelines for reference lists.  Some titles are fully capitalized, while others have only the first word of the title and proper nouns capitalized. You can manage this in Endnote by without having to change the titles of references by editing the output settings. In the bibliography section choose sentence style capitalization in the title capitalization subsection.  Although, recheck the titles carefully as proper nouns such as place names will have had their capitalization removed.

https://journals.plos.org/plosone/s/submission-guidelines#loc-references

We look forward to receiving your revised manuscript.

Kind regards,

Jane Anne Scott, PhD, MPH Grad Dip Dietetics, BSc

Academic Editor

PLOS ONE

Additional Editor Comments:

Minor corrections

Line 60 Other than when used in reference to grains, i.e. intact grains, the phrase ‘intact foods’ is not commonly used to describe other groups of unprocessed or minimally processed foods.

Lines 69 to 70. This sentence referring to access to ultra-processed foods is not needed in this paragraph justifying the FOPL. You have already linked ultra-processed foods to NCDs in lines 56-58.

Line 129 and elsewhere, data is the plural of datum so should be ‘data were’ collected.

Lines 211-212 presumably the moderator first transcribed the recordings verbatim and THEN translated the data into English where applicable. In which case this sentence should be reordered.

Lines 227-228 The aim of the study does not need to be repeated here.

Line 501 the use of the word thus in this sentence implies that the reason for the preference for black is explained in the first half of the sentence, when in fact the rationale for preferring black is provided in the second half of the sentence. Therefore, suggest rewording as ‘black was deemed more effective’.

Line 511 presumably you mean a white triangle on a black background and NOT a black triangle on a black background

Line 543 should read ultra-processed foods (plural)

Line 544 should read labels (plural)

Journal Requirements:

2.  When reporting the results of qualitative research, we suggest consulting the COREQ guidelines or other relevant checklists listed by the Equator Network, such as the SRQR, to ensure complete reporting (http://journals.plos.org/plosone/s/submission-guidelines#loc-qualitative-research). In this case, please consider including more information on the number of interviewers, their training and characteristics. Moreover, please provide the interview guide used as a Supplementary file.

[We thank School of Public Health at the University of the Western Cape and the DSI/NRF CoE in Food Security UID 91490) for support.]

 [This study was funded by Bloomberg Philanthropies. The funders had no role in study design, data collection and analysis, decision to publish or preparation of the manuscript].

4. Thank you for stating the following in the Financial Disclosure section:

[This study was funded by Bloomberg Philanthropies. The funders had no role in study design, data collection and analysis, decision to publish or preparation of the manuscript].

We note that you received funding from a commercial source: Bloomberg L.P.

6. We note that Figures in your submission contain copyrighted images. All PLOS content is published under the Creative Commons Attribution License (CC BY 4.0), which means that the manuscript, images, and Supporting Information files will be freely available online, and any third party is permitted to access, download, copy, distribute, and use these materials in any way, even commercially, with proper attribution. For more information, see our copyright guidelines: http://journals.plos.org/plosone/s/licenses-and-copyright.

1.         You may seek permission from the original copyright holder of Figures to publish the content specifically under the CC BY 4.0 license.

Reviewers' comments:

Reviewer's Responses to Questions

**Comments to the Author**

1. Is the manuscript technically sound, and do the data support the conclusions?

Reviewer #1: Partly

Reviewer #2: Partly

2. Has the statistical analysis been performed appropriately and rigorously? 

Reviewer #1: Yes

Reviewer #2: N/A

3. Have the authors made all data underlying the findings in their manuscript fully available?

Reviewer #1: Yes

Reviewer #2: No

4. Is the manuscript presented in an intelligible fashion and written in standard English?

Reviewer #1: Yes

Reviewer #2: Yes

5. Review Comments to the Author

Reviewer #1: Thank you for letting me review this manuscript. This is a topic worthy of discussion since the high prevalence of overweight and obesity. However, there are some comments:

1. In line 150, please add the references after the sentence: "...detailed design brief based on the latest literature"

2. According to the following article (An, 2021), Graphic with health effect labels showed the largest impact on dissuading consumers from choosing them. Why the design of graphic (e.g., graph of health effect, which displays a picture of an obese belly or decayed teeth with relevant descriptions, and graphic with nutrient profile, which displays a picture of sugar added in the drinks with corresponding descriptions) was not used in the label design in this study?

An R, Liu J, Liu R, Barker AR, Figueroa RB, McBride TD. Impact of Sugar-Sweetened Beverage Warning Labels on Consumer Behaviors: A Systematic Review and Meta-Analysis. Am J Prev Med. 2021;60(1):115-126. doi:10.1016/j.amepre.2020.07.003

3. There are lots of mistakes in the reference, please revise the format of the references.

Reviewer #2: The manuscript deals with a relevant topic for public health policy worldwide. The manuscript addresses South African citizens’s perception of warning labels. However, major changes are needed before the manuscript can be considered for publication. Details comments are provided below.

Introduction

- The introduction fails to provide a comprehensive overview of the topic. The authors did not include a thorough analysis of the literature to clearly convey what is known about the topic and what are the knowledge gaps. The authors should include a more detailed analysis of the growing body of evidence around warning labels.

- Considering that the manuscript is focused on warning labels, details on other FOP nutrition labelling schemes is not necessary. The inclusion of Figure 1 is not necessary in the context of the manuscript.

- The contribution of the manuscript should be more clearly presented. How does the manuscript contribute to the literature? Is the contribution related to the specific context (South Africa)?

Objectives

- The authors state that one of the aims of the manuscript was to identify “features that enhance or diminish the effectiveness of a warning label”. However, the design is not appropriate to address this objective. Qualitative research enables to explore a specific topic but cannot be regarded as appropriate to evaluate the effectiveness of a public policy or to identify features that enhance or diminish its effectiveness. The authors could refer to “identify citizens’ views on features that could influence the effectiveness of warnings” or something like that.

Materials and Methods

- How was the number of participants selected?

- Did the authors rely on theoretical sampling for the design of the study?

- How did the authors manage participants’ heterogeneity? Did they conduct focus groups with participants from very different settings? This should be better explained.

- What was the purpose of creating so many warning labels? Do the authors think that participants were actually able to pay attention to the nuances between all the designs?

- Qualitative research is not appropriate to select the most attention grabbing, the most effective or the most likely to influence purchasing behavior. Quantitative research should have been used for this purpose. In addition, the social interactions during focus groups make it not possible to assess individual opinions on the topic.

- The authors should have included the question guide. It is an essential element to evaluate the validity and reliability of qualitative research.

- How did the authors handle the influence of participants’ characteristics on their opinions?

Results

- Additional details are needed in the Results section to more clearly convey the results. I recommend the authors to include quotes in Table 3.

- The authors should be careful about the interpretation of the results. As I have previously mentioned, they are dealing with focus groups and therefore results should be interpreted considering their qualitative nature. Several changes should be made throughout the text.

- I recommend the authors not to refer to “the most understood design features”.

- How could the authors generalize their results to different groups of participants? For example, they stated that “participants from all socio-economic backgrounds” had positive attitudes towards the warnings. They did conduct separate focus groups? Did all participants explicitly stated their opinion?

Discussion

- The flow and clarity of the Discussion section could be improved.

- The authors should rewrite many parts of the discussion where they refer to the comparison of labels and design features considering the type of data they are dealing with. For example, they state “black was more effective”.

- An important point the authors should discuss is related to the contraposition between citizens’ opinions and actual effectiveness of the policy. This is particularly relevant for the discussion, as the authors could encourage other researchers to base policy decisions exclusively on qualitative data.

- As far as I know, the Uruguay nutrition labelling policy entered into force in 2020, so I guess that there are no studies showing changes in expenditure. This should be clarified in line 527, as I think the authors are mixing up an experimental study with the evaluation of policy impact.

- The limitations of the study should be better acknowledged in the paper, as well as suggestions for further research.

6. PLOS authors have the option to publish the peer review history of their article (what does this mean?). If published, this will include your full peer review and any attached files.

Reviewer #1: **Yes: **Jianxiu Liu

Reviewer #2: No

---

## [Author Response · Author response to Decision Letter 0]

5 Jul 2021

1. In particular, as recommended by reviewer 2, please include a more comprehensive overview of the use of warning labels in your introduction. For instance, you do not describe pictorial or graphic warning labels as described by Pechey et al. While this publication postdates the conduct of the focus groups, this style of warning label should nevertheless be described and the relative merits discussed. 

Our response: Thank you for the reference to the paper by Pechey et al. (2020). This article was considered as a model for reviewing the introduction of our paper. 

A more comprehensive overview of warning labels is now included. More information regarding design elements that improve the efficacy of warning labels is included as well as discussion of emerging studies around warning labels - lines 105-142

2. Lines 488-498 I think more could be made of the differences between countries regarding consumer shape preferences, as justification for the need to replicate this type of research locally. This finding highlights the importance of having culturally relevant shapes, as shapes are not universally interpreted in the same manner. 

Our response: Thank you for this comment. We now refer to the fact that shapes are not interpreted the same way and that each country should explore shape preferences within its own context - Lines 588-597

3. Please check the guidelines for reference lists. Some titles are fully capitalized, while others have only the first word of the title and proper nouns capitalized. You can manage this in Endnote by without having to change the titles of references by editing the output settings. In the bibliography section choose sentence style capitalization in the title capitalization subsection. Although, recheck the titles carefully as proper nouns such as place names will have had their capitalization removed.

https://journals.plos.org/plosone/s/submission-guidelines#loc-references

Our response: Thank you for pointing out aspects where guidelines were not applied correctly. We have now attended to formatting and all titles are in sentence case. 

Minor corrections

4. Line 60 Other than when used in reference to grains, i.e. intact grains, the phrase ‘intact foods’ is not commonly used to describe other groups of unprocessed or minimally processed foods. 

Our response: Thank you for this comment. We have now removed the word ‘intact’ and substituted it with ‘whole’– line 60

5. Lines 69 to 70. This sentence referring to access to ultra-processed foods is not needed in this paragraph justifying the FOPL. You have already linked ultra-processed foods to NCDs in lines 56-58. 

Our response: Thank you for noting that the sentence is not needed. We have now deleted the sentence - lines 69-70

6. Line 129 and elsewhere, data is the plural of datum so should be ‘data were’ collected. 

Our response: Thank you for the comment. We have edited the manuscript and the phrase “data was” is now edited to “data were” throughout the manuscript. 

7. Lines 211-212 presumably the moderator first transcribed the recordings verbatim and THEN translated the data into English where applicable. In which case this sentence should be reordered. 

Our response: Thank you for this comment. We have reordered the sentence – lines 272-273

8. Lines 227-228 The aim of the study does not need to be repeated here. 

Our response: Thank you for the suggestion. The aim of the study was deleted – lines 292 - 293

9. Line 501 the use of the word thus in this sentence implies that the reason for the preference for black is explained in the first half of the sentence, when in fact the rationale for preferring black is provided in the second half of the sentence. Therefore, suggest rewording as ‘black was deemed more effective’. 

Our response: We substituted the term ‘thus’ with ‘deemed’ – line 600 - 601

10. Line 511 presumably you mean a white triangle on a black background and NOT a black triangle on a black background, 

Our response: Thank you for the comment on the framing of the sentence in line 511. We have left the sentence as it is as a black triangle was also tested against a black background - line 612

11. Line 543 should read ultra-processed foods (plural) 

Our response: Thank you for noting the need to change the word into plural. We have now changed the word ‘food’ to ‘foods’ - line 646

12. Line 544 should read labels (plural) 

Our response: Thank you for the comment. We have now changed the term ‘label’ to ‘labels’ - line 648

Our response: Thank you for the comment and for referring us to these websites. We have done the following:

• Inserted pilcrow to indicate 1st set of equal contributors (cover page)

• Corrected punctuation after each author’s name (cover page)

• Abbreviation (NY) removed under affiliations (cover page)

• Corrected punctuation when labelling Figures (e.g. Substituted colon with a full stop after Fig 1 and Fig 3 when labelling Figures)

• We have attempted to label the files correctly.

13. When reporting the results of qualitative research, we suggest consulting the COREQ guidelines or other relevant checklists listed by the Equator Network, such as the SRQR, to ensure complete reporting (http://journals.plos.org/plosone/s/submission-guidelines#loc-qualitative-research). In this case, please consider including more information on the number of interviewers, their training and characteristics. Moreover, please provide the interview guide used as a Supplementary file.

Our response: Thank you for the comments and the suggestion to consult the COREX guidelines regarding reporting of the qualitative research results. We consulted the COREX guidelines and added the following:

14. Consider including more information on the number of interviewers, their training and characteristics. 

We have now indicated that all the interviews were conducted by one moderator - line 243 and that the moderator had extensive experience in qualitative data collection and data analysis - line 244-245

15. Please provide the interview guide used as a Supplementary file

We have now attached the focus group discussion guide or moderator guide as a Supplementary File (S2 File).

16. Thank you for stating the following in the Acknowledgments Section of your manuscript: [We thank School of Public Health at the University of the Western Cape and the DSI/NRF CoE in Food Security UID 91490) for support.]

 [This study was funded by Bloomberg Philanthropies. The funders had no role in study design, data collection and analysis, decision to publish or preparation of the manuscript].

Our response: Thank you for the opportunity to clarify this point. The authors did not receive any funding from the DSI/NRF CoE in Food Security, but only administrative support – line 673. 

Our Funding Statement therefore remains: [This study was funded by Bloomberg Philanthropies. The funders had no role in study design, data collection and analysis, decision to publish or preparation of the manuscript].

17. Thank you for stating the following in the Financial Disclosure section: [This study was funded by Bloomberg Philanthropies. The funders had no role in study design, data collection and analysis, decision to publish or preparation of the manuscript].

We note that you received funding from a commercial source: Bloomberg L.P. Please provide an amended Competing Interests Statement that explicitly states this commercial funder, along with any other relevant declarations relating to employment, consultancy, patents, products in development, marketed products, etc.

Our response. Thank you for noting the mentioning of Bloomberg LP as a funder in this study. We would like to draw it to your attention that Bloomberg LP was erroneously mentioned and the study was actually funded by Bloomberg Philanthropies, as explained in the cover letter. 

18. We note that you have stated that you will provide repository information for your data at acceptance. Should your manuscript be accepted for publication, we will hold it until you provide the relevant accession numbers or DOIs necessary to access your data. If you wish to make changes to your Data Availability statement, please describe these changes in your cover letter and we will update your Data Availability statement to reflect the information you provide.

Our response: This is noted and there are no changes to the statement

19. We note that Figures in your submission contain copyrighted images. All PLOS content is published under the Creative Commons Attribution License (CC BY 4.0), which means that the manuscript, images, and Supporting Information files will be freely available online, and any third party is permitted to access, download, copy, distribute, and use these materials in any way, even commercially, with proper attribution. For more information, see our copyright guidelines: http://journals.plos.org/plosone/s/licenses-and-copyright. 

Our response: Thank you for the comment. We removed all images with copyrights and only remain with images that can be freely available online, and can be used in any way, with proper attribution.

20. Reviewer #1: Thank you for letting me review this manuscript. This is a topic worthy of discussion since the high prevalence of overweight and obesity. However, there are some comments:

In line 150, please add the references after the sentence: "...detailed design brief based on the latest literature" 

Our response: We have now cited sources of the design brief submitted to the designers - line 208

22. According to the following article (An, 2021), Graphic with health effect labels showed the largest impact on dissuading consumers from choosing them. Why the design of graphic (e.g., graph of health effect, which displays a picture of an obese belly or decayed teeth with relevant descriptions, and graphic with nutrient profile, which displays a picture of sugar added in the drinks with corresponding descriptions) was not used in the label design in this study? 

An R, Liu J, Liu R, Barker AR, Figueroa RB, McBride TD. Impact of Sugar-Sweetened Beverage Warning Labels on Consumer Behaviors: A Systematic Review and Meta-Analysis. Am J Prev Med. 2021;60(1):115-126. doi:10.1016/j.amepre.2020.07.003 

Our response: Thank you for the comment and reference to an article by An et al. (2020). We have considered different warning label formats and noted that in literature warning labels containing familiar shapes such as an octagon or a triangle improved consumers understanding of nutrient composition and assisted consumers to identify unhealthy products.

Introduction

23. The introduction fails to provide a comprehensive overview of the topic. The authors did not include a thorough analysis of the literature to clearly convey what is known about the topic and what are the knowledge gaps. The authors should include a more detailed analysis of the growing body of evidence around warning labels. 

Our response: Thank you for the suggestion to improve on the introduction. We have now provided more information about warning labels and interrogated the growing evidence around them, including their merits – lines 105 - 142

24. Considering that the manuscript is focused on warning labels, details on other FOP nutrition labelling schemes is not necessary. The inclusion of Figure 1 is not necessary in the context of the manuscript. 

Our response: Thank you for the comment: We have deleted Fig 1 (lines 86 and 87) and have removed detailed references to either the GDA or the MTL from the introduction (lines 88-94)

25. The contribution of the manuscript should be more clearly presented. How does the manuscript contribute to the literature? Is the contribution related to the specific context (South Africa)? 

Our response: Thank you for pointing out the failure of the manuscript to clearly indicate its contribution. This study aims to explore the citizens’ view of warning labels as a potential labelling format to guide food purchasing within the South African context. We revised the manuscript to capture that – lines 151 - 164

Objectives

26. The authors state that one of the aims of the manuscript was to identify “features that enhance or diminish the effectiveness of a warning label”. However, the design is not appropriate to address this objective. Qualitative research enables to explore a specific topic but cannot be regarded as appropriate to evaluate the effectiveness of a public policy or to identify features that enhance or diminish its effectiveness. The authors could refer to “identify citizens’ views on features that could influence the effectiveness of warnings” or something like that. 

Our response: Thank you for the comment. We amended the objective to reflect that the aim of the study was to explore citizens’ views on features that could influence the effectiveness of warnings, instead of identifying design features that enhance or diminish the effectiveness of warnings - line 160-164

Materials and Methods

27. How was the number of participants selected? 

Our response: The total number targeted was 120, 40 from each province and they were selected to reach a quota for socio-economic status, urbanicity vs. rurality, gender and age Lines 184-191. 

28. Did the authors rely on theoretical sampling for the design of the study? 

Our response: No, we used purposive sampling where we decided on the criteria and recruited all the participants before we commenced with data collection.

29. How did the authors manage participants’ heterogeneity? Did they conduct focus groups with participants from very different settings? This should be better explained. 

Our response. Thank you for the question. All the focus groups were homogenous, each group consisted of participants with similar sociodemographic characteristics - lines 189-191

30. What was the purpose of creating so many warning labels? Do the authors think that participants were actually able to pay attention to the nuances between all the designs?

Our response: The aim was for participants to choose from a wide range of elements and not to make the choice options too limited. We refer to a study where a number of prototypes were shown to the participants - line 237-238. 

31. Qualitative research is not appropriate to select the most attention grabbing, the most effective or the most likely to influence purchasing behaviour. Quantitative research should have been used for this purpose. In addition, the social interactions during focus groups make it not possible to assess individual opinions on the topic.

Our response: We revised the manuscript to capture that the responses reflect opinions of the group in general and all reference pointing to the study determining the ‘most attention grabbing, the most effective or the most likely to influence purchasing behaviour has been removed throughout the manuscript – lines 161-162; 230; 267 – 269; 547-549.

32. The authors should have included the question guide. It is an essential element to evaluate the validity and reliability of qualitative research. 

Our response: Thank you for suggesting that we include the interview guide. We have now attached it as a Supplementary File (S2 File)

33. How did the authors handle the influence of participants’ characteristics on their opinions? 

Our response: We grouped participants according to their various characteristics (age, literacy level and economic status) to allow maximum participation. 

Results

34. Additional details are needed in the Results section to more clearly convey the results. I recommend the authors to include quotes in Table 3. 

Our response: We have attached themes, subthemes and quotes as Supplementary Table (S1 Table)

35. The authors should be careful about the interpretation of the results. As I have previously mentioned, they are dealing with focus groups and therefore results should be interpreted considering their qualitative nature. Several changes should be made throughout the text. 

Our response: Thank you for the comment. We revised the entire results section and the abstract and amended to suit qualitative results. 

36. I recommend the authors not to refer to “the most understood design features”. 

Our response: We changed the heading to: Elements perceived effective as warning - line 545

37. How could the authors generalize their results to different groups of participants? For example, they stated that “participants from all socio-economic backgrounds” had positive attitudes towards the warnings. They did conduct separate focus groups? Did all participants explicitly stated their opinion? 

Our response: Thank you for the opportunity to clarify. We grouped participants according to their sociodemographic status and analysed responses per focus group. The warning labels were positively appraised in all the focus groups 

Discussion

38. The flow and clarity of the Discussion section could be improved. 

Our response: We revised the entire discussion section and attempted to improve on its flow and clarity. 

39. The authors should rewrite many parts of the discussion where they refer to the comparison of labels and design features considering the type of data they are dealing with. For example, they state “black was more effective”. 

Our response: We rewrote the discussion section to reflect the qualitative nature of the study. For example, we refrained from depicting some warning elements as being more effective than others and instead expressed participants’ views about those elements – lines 602-603; 611-612

40. An important point the authors should discuss is related to the contraposition between citizens’ opinions and actual effectiveness of the policy. This is particularly relevant for the discussion, as the authors could encourage other researchers to base policy decisions exclusively on qualitative data. 

Our response: Any contrapositions made between participants views and policy were reviewed and sentences constructed differently - lines 41-42; 409, 600-601; 651)

40. As far as I know, the Uruguay nutrition labelling policy entered into force in 2020, so I guess that there are no studies showing changes in expenditure. This should be clarified in line 527, as I think the authors are mixing up an experimental study with the evaluation of policy impact. 

Our response: Thank you for the comment. We have now indicated that the studies were experimental - lines 629 and 640

41. The limitations of the study should be better acknowledged in the paper, as well as suggestions for further research.

Our response: Thank you for the suggestion to better acknowledge the limitations of the study and make suggestions for future research. We have expanded on the limitation of the study (lines 659-660) and made suggestions for future research (lines 661-662).

---

## [Decision Letter · Decision Letter 1]

7 Sep 2021

The Editor

South African consumers' perceptions of front-of-package warning labels on unhealthy foods and drinks

PONE-D-21-13158R1

Dear Dr. Bopape,

We’re pleased to inform you that your manuscript has been judged scientifically suitable for publication and will be formally accepted for publication once it meets all outstanding technical requirements.

Kind regards,

Jane Anne Scott, PhD, MPH Grad Dip Dietetics, BSc

Academic Editor

PLOS ONE

Additional Editor Comments (optional):

Reviewers' comments:

Reviewer's Responses to Questions

**Comments to the Author**

1. If the authors have adequately addressed your comments raised in a previous round of review and you feel that this manuscript is now acceptable for publication, you may indicate that here to bypass the “Comments to the Author” section, enter your conflict of interest statement in the “Confidential to Editor” section, and submit your "Accept" recommendation.

Reviewer #1: All comments have been addressed

Reviewer #2: All comments have been addressed

2. Is the manuscript technically sound, and do the data support the conclusions?

Reviewer #1: Yes

Reviewer #2: Yes

3. Has the statistical analysis been performed appropriately and rigorously? 

Reviewer #1: Yes

Reviewer #2: N/A

4. Have the authors made all data underlying the findings in their manuscript fully available?

Reviewer #1: Yes

Reviewer #2: Yes

5. Is the manuscript presented in an intelligible fashion and written in standard English?

Reviewer #1: Yes

Reviewer #2: Yes

6. Review Comments to the Author

Reviewer #1: The study explores consumers’ perceptions of warning labels and consumers’ views on design features that could influence the effectiveness of a warning label. After the first round of revising, the manuscript improved and can be accepted now.

Reviewer #2: Thank you for addressing all my comments. The manuscript has improved and, in my opinion, can be accepted for publicaiton.

7. PLOS authors have the option to publish the peer review history of their article (what does this mean?). If published, this will include your full peer review and any attached files.

Reviewer #1: No

Reviewer #2: No

---

## [Editor Report · Acceptance letter]

17 Sep 2021

PONE-D-21-13158R1 

South African consumers’ perceptions of front-of-package warning labels on unhealthy foods and drinks 

Dear Dr. Bopape:

I'm pleased to inform you that your manuscript has been deemed suitable for publication in PLOS ONE. Congratulations! Your manuscript is now with our production department. 

Kind regards, 

on behalf of

Dr. Jane Anne Scott 

Academic Editor

PLOS ONE